# Intimate Relationships and Stroke: Piloting a Dyadic Intervention to Improve Depression

**DOI:** 10.3390/ijerph19031804

**Published:** 2022-02-05

**Authors:** Alexandra L. Terrill, Maija Reblin, Justin J. MacKenzie, Brian R. W. Baucom, Jackie Einerson, Beth Cardell, Lorie G. Richards, Jennifer J. Majersik

**Affiliations:** 1Department of Occupational & Recreational Therapies, University of Utah, Salt Lake City, UT 84108, USA; jackie.einerson@hsc.utah.edu (J.E.); beth.cardell@hsc.utah.edu (B.C.); lorie.richards@hsc.utah.edu (L.G.R.); 2Department of Family Medicine, University of Vermont, Burlington, VT 05405, USA; maija.reblin@med.uvm.edu; 3Division of Physical Medicine & Rehabilitation, University of Utah, Salt Lake City, UT 84132, USA; Justin.mackenzie@hsc.utah.edu; 4Department of Psychology, University of Utah, Salt Lake City, UT 84112, USA; brian.Baucom@utah.edu; 5Department of Neurology, University of Utah, Salt Lake City, UT 84132, USA; jennifer.majersik@hsc.utah.edu

**Keywords:** stroke, post-stroke depression, depression, caregiver, dyadic intervention, positive psychology

## Abstract

Stroke affects not only the survivor but also their romantic partner. Post-stroke depression is common in both partners and can have significant negative consequences, yet few effective interventions are available. The purpose of this study was to pilot test a novel 8-week remotely administered dyadic intervention (ReStoreD) designed to help couples better cope with stroke-related changes and reduce depressive symptoms. Thirty-four cohabitating survivor–partner dyads at least 3 months post-stroke and reporting some changes in mood were enrolled. Depressive symptoms were assessed pre- and post-intervention and at 3-month follow-up. Repeated measures analysis of variance was used to assess the effects of ReStoreD over time on depressive symptoms in stroke survivors and their partners. Twenty-six dyads completed the study. Although statistical significance was not reached, there was a large effect size for improvements in depressive symptoms for stroke survivors. There was no significant improvement for partners, and the effect size was minimal. Those with more significant depressive symptoms at baseline were more likely to benefit from the intervention. This pilot study established proof-of-concept by demonstrating that depressive symptoms can be lessened in stroke survivors and partners with more severe depressive symptoms. Future research will establish the efficacy of the intervention in a fully powered study.

## 1. Introduction

One-third of persons with stroke experience post-stroke depression (PSD), characterized by low mood, decreased energy, inability to feel pleasure in normally pleasurable activities (anhedonia), fatigue, and changes in appetite, concentration, and sleep [1]. The etiology of PSD is not well understood [2], though it is likely a combination of the physical and neurocognitive effects of the stroke itself, in addition to the psychological and social effects of a chronic medical condition. PSD is associated with significant negative consequences and outcomes, including increased disability and mortality, decreased participation, and poorer quality of life [3,4,5,6].

Depression also occurs in up to 60% of stroke care partners [7]. Care partners, often a spouse/partner [8], provide emotional support to the person with stroke and help with symptom management and/or other care tasks. Care partner depression can contribute to social isolation and declines in their own personal health [9], as well as interfere with rehabilitation and increase the likelihood of re-hospitalization of the person with stroke [10]. Research has shown that well-being is interdependent in couples, meaning that if one partner is depressed, the other is more likely to also be depressed [11,12]. As such, fostering well-being in both partners is important for optimizing short- and long-term health outcomes for both persons with stroke and their care partners. Despite the high prevalence and significant consequences of mental health issues post-stroke, treatment available to persons with stroke and care partners is often inadequate [1]. 

To address the mental health needs of both persons with stroke and their care partners, it may be beneficial to use a couples-based (dyadic) treatment approach [13,14], in which both partners in the dyad are active participants in the intervention. Prior dyadic interventions primarily targeted the individual with stroke and are insufficient to meet the needs of the couple as a whole. To meet these needs, we developed an 8-week dyadic positive psychology-based intervention (PPI), called ReStoreD (Resilience after Stroke in Dyads) [15]. Interventions based in positive psychology offer a re-orientation to supplement the traditional “fix-what’s-wrong” approach and seek to build on individuals’ strengths, resources, values, and hopes to increase overall well-being [16]. As such, PPIs are ideally suited for populations with chronic medical conditions and/or disability, as they emphasize existing and preserved strengths and assets within the context of severe and sometimes permanent changes in function and independence [17]. Though we are among the first to apply PPIs to a stroke population, they have been effectively applied to other medical populations [18,19,20] and have been shown to significantly increase well-being and decrease depressive symptoms [16,21,22] with long-lasting effects [23]. Though typically delivered at the individual level, PPIs delivered to a dyad have the potential to not only improve depressive symptoms in the individual but also have a synergistic effect between the care partner and the person with stroke.

The purpose of this study was to pilot test ReStoreD in couples coping with stroke to determine preliminary effects on depressive symptoms. We hypothesized that participating in the intervention would significantly improve depressive symptoms in both persons with stroke and care partners, and that improvements would be maintained at the 3-month follow-up. Further, the Actor–Partner Interdependence Model (APIM [24]) was used to explore potential reciprocal effects of depressive symptoms between dyad members.

## 2. Materials and Methods

### 2.1. Design

The pilot study used a randomized waitlist-controlled design to test the effects of ReStoreD on reducing depressive symptoms in couples coping with stroke. See Figure 1 for enrollment data. Block randomization was used to assign 60% of the dyads to immediately receive the intervention and 40% to the waitlist condition. However, due to issues with attrition (unrelated to the intervention and documented in Figure 1) in the waitlist group, and because no significant baseline differences were found between groups, analysis of the entire sample was conducted using a single-arm repeated measures analysis of variance (ANOVA) of pre- and post-intervention and 3-month follow-up assessments. The study was pre-registered at ClinicalTrials.gov (NCT#03335358). The data that support the findings of this study are available to additional investigators from the corresponding author upon reasonable request.

### 2.2. Participants

Consistent with recommendations for pilot study sample sizes [25], the target sample size for the planned analysis was 24 dyads; a target total sample size of 34 dyads was planned to account for attrition. This sample size was expected to be sufficient to demonstrate proof-of-concept as long as changes were in the expected direction. Participant dyads consisted of one partner who had a stroke at least 3 months prior to enrolling in the study and a partner or spouse. Participants were recruited through referrals from University of Utah-affiliated outpatient rehabilitation and neurology clinics and in-person at community-based events. Dyads meeting the following criteria were eligible: (1) 18 years of age or older; (2) living together for at least 6 months; (3) community-dwelling; and (4) either one or both partner(s) had to report changes in mood (depressed mood and/or anhedonia) since the stroke as assessed by self-report on the 2-item Patient Health Questionnaire [26]. A clinical diagnosis of depression was not required. Dyads were excluded if: (1) they were unable to understand printed English instructions, (2) either partner did not want to participate, (3) the care partner had a history of stroke or other major neurologic condition, and/or (4) the person with stroke had significant cognitive impairment or aphasia that would interfere with meaningful participation in the intervention. Although individuals with severe cognitive impairment and/or aphasia were excluded, those with mild–moderate cognitive impairment (assessed using the Montreal Cognitive Assessment (MoCA) [27] or the Cognitive Assessment for Stroke Patients (CASP) [28]) and/or expressive aphasia were included in the study to increase generalizability to a stroke population [29,30]. The University of Utah Institutional Review Board approved all study procedures prior to recruitment and data collection. All participants provided their own written informed consent.

### 2.3. Procedures

All participants completed self-report assessments pre- (T1), immediate post- (T2), and 3 months post-intervention (T3) in person at a University of Utah-affiliated clinic. Two trained research assistants administered assessments at each session. All data were entered and maintained in REDCap (Research Electronic Data Capture tools [31]), a highly secure, web-based application designed to support data capture. An investigator who did not conduct assessments conducted statistical analysis.

#### Intervention

Dyads received a manualized 20-min training from trained staff on how to complete the 8-week self-administered intervention at the end of the T1 visit. Each participant received their own activity booklet containing instructions and a tracking calendar to log weekly activities at home. Participants could choose from a variety of activities (see Table 1) each week and were asked to complete at least two individual activities and two couples’ activities for at least 15 min each (total time = 1 h per week). Specific activities in the ReStoreD intervention were selected based on empirically supported evidence for efficacy in improving depressive symptoms and/or increasing well-being [21,22]. Research assistants completed protocolized weekly check-in phone calls to remind participants to complete their activities and answer any questions. Protocol fidelity was assessed throughout the study.

### 2.4. Measures

Basic demographic and stroke-related information was collected at enrollment. The primary outcome was assessed at each time point using the PROMIS^®^ (Patient-Reported Outcomes Measurement Information System) Depression Short Form v.8b (PROMIS-D-SF), which is an 8-item psychometrically sound self-report instrument that has been developed and validated for use with the general population and with individuals living with chronic conditions such as stroke [32], and it is responsive to change [33]. Participants rated items such as “In the past 7 days, I felt depressed” on a 5-point scale, with 1 = never and 5 = always. Scores range from 8 to 40 points, with higher scores indicating more severe depressive symptoms. Raw scores were used in our analyses.

### 2.5. Statistical Analysis

All dyads were included as part of a single analysis. Repeated measures ANOVA using three time points (T1, T2, and T3) and partner status (person with stroke vs. care partner) was used to examine effects on depressive symptoms over time and to provide preliminary effect sizes. Exploratory post hoc analyses were conducted to identify subgroups who are most likely to benefit from the intervention (younger vs. older couples coping with stroke (based on young stroke definition of 55 years old or younger [34]), men vs. women, and affected cerebral hemisphere). Additional sensitivity analyses for these effects were tested using multilevel modeling (MLM). MLM is a recommended method for conducting intent-to-treat analyses using all available data. The magnitude, direction, and significance level of the results of MLMs were highly similar to those produced by ANOVAs [35].

Finally, an Actor–Partner Interdependence Model (APIM) was used to estimate the association between dyad members’ scores at each time point (interdependence effects). The model examines direct and indirect effects for how each partner’s T1 PROMIS-D-SF score predicts their own (actor effects) as well as their partner’s (partner effects) T2 and T3 scores. These models were estimated using a Bayesian structural equation model; this produces stable and unbiased parameter estimates with sample sizes *n* = 30 or larger [36].

## 3. Results

### 3.1. Participant Descriptive Data

Study participants consisted of 34 cohabitating partner dyads (*n* = 68). Both partners with stroke and care partners had a mean age of 53 years, but fewer partners with stroke were female (41%) than care partners (59%). The mean length of relationship was 25 years, and 91% were married. See Table 2 for additional participant characteristics. Other than gender, basic demographic information for persons with stroke and care partners was similar at baseline. A range of stroke types and locations were represented among participants with stroke. Consistent with inclusion criteria, cognition screening scores of individuals with stroke were indicative of a range of impairment, from none to moderate. A range of physical function was also represented, as indicated by scores on the Lawton Instrumental Activities of Daily Living (IADL) scale administered pre-intervention (range of 9–24; mean = 18.83, SD = 4.44).

At enrollment, individuals with stroke had an average PROMIS-D-SF score of 17.09 (SD = 6.93), and care partners had an average score of 14.12 (SD = 5.53; t(64) = 1.92, *p* = 0.06). Subgroup analyses show that care partners who were 55 years old or younger had non-significantly higher baseline PROMIS-D-SF scores (M = 15.65, SD = 6.33) compared to those over 55 years old (M = 12.50, SD = 4.13; t(31) = 1.68, *p* = 0.10). Similarly, there was no significant difference in baseline PROMIS-D-SF scores in persons with stroke 55 years old or younger (M = 18.06, SD = 7.57) compared to those over 55 (M = 16.18, SD = 6.37). Among persons with stroke, those with right hemisphere involvement reported higher PROMIS-D-SF scores (M = 18.10, SD = 5.59) as compared to those with left hemisphere (M = 17.35, SD = 7.80) or other (M = 14.67, SD = 6.93) involvement, though these differences were not statistically significant (F(2, 30) = 0.47, *p* = 0.63). There was no relationship between time since stroke and PROMIS-D-SF scores at baseline for persons with stroke (r = −0.10, *p* = 0.58) or care partners (r = 0.01, *p* = 0.94).

### 3.2. Intervention Effects

To examine the effects of the intervention on partners with stroke and care partners, two separate one-way repeated measures ANOVAs were conducted to evaluate changes in participants’ PROMIS-D-SF scores at T1, T2, and T3. Interpretations of effect sizes are based on Cohen’s rule-of-thumb for eta-squared (η^2^): 0.01 = small; 0.06 = medium; and >0.14 = large effect size [37]. For the person with stroke, the results indicated a marginally non-significant time effect for improvements in depressive symptoms (Wilks’ Lambda = 0.78, F(2, 23) = 3.23, *p* = 0.06). Although not statistically significant, this suggests a large intervention effect size (η^2^ = 0.22) on PROMIS-D-SF scores [38]. For care partners, the results of the analysis indicated no significant time effect (Wilks’ Lambda = 0.99, F(2, 22) = 0.01, *p* = 0.99); i.e., the effect size did not vary between T1, T2, and T3, and the effect size was minimal (η^2^ = 0.01). Table 3 shows mean PROMIS-D-SF scores and standard deviations for persons with stroke and care partners at T1, T2, and T3. Individuals with stroke on average had a significant decrease in PROMIS-D-SF scores from T1 to T2. An exploratory post hoc analysis of those who scored in the 50th percentile or higher on the PROMIS-D-SF (≥14 for partners with stroke, ≥12 for care partners) at T1 showed a significant time effect for partners with stroke (Wilks’ Lambda = 0.58, F(2, 11) = 4.07, *p* < 0.05), with a large effect size (η^2^ = 0.43) [37]. Time effects for more depressed care partners in this exploratory analysis remained non-significant (Wilks’ Lambda = 0.86, F(2, 11) = 0.90, *p* = 0.43), but with a medium effect size (η^2^ = 0.14).

### 3.3. APIM Analysis

An APIM was estimated using Bayesian structural equation modeling to explore the relationship between the scores of members of a dyad at each time point. Figure 2 provides a simplified path model, and Table 4 shows all direct, specific indirect, and total effects. The results show significant actor effects for the individual with stroke in that their PROMIS-D-SF score at T1 significantly predicted their own PROMIS-D-SF score at T2 but not T3. The care partner’s PROMIS-D-SF score at T1 significantly predicted their own PROMIS-D-SF score at T2, and T3 was significantly predicted by their scores at T1 and T2.

There were no significant partner effects, i.e., neither partner’s PROMIS-D-SF scores were significantly associated with the other partner’s scores. However, there was near significance (*p* < 0.06) for PROMIS-D-SF scores of the individual with stroke at T1 predicting PROMIS-D-SF scores of the care partner at T3, with higher scores for individuals with stroke at T1 predicting lower care partner scores at T3. The indirect pathways through T2 scores were not significant.

## 4. Discussion

Depression is common in both persons who had a stroke and their care partners. A growing body of evidence supports a biopsychosocial model for both etiology and treatment approaches [12,13,14,39], and recent recommendations highlight the need for interventions supporting the psychosocial needs of stroke survivor–care partner dyads [12,13,14]. This study tested the effects of a novel dyadic positive psychology intervention, ReStoreD, on depressive symptoms in persons with stroke and their care partners. Our findings suggest that ReStoreD may reduce depressive symptoms in persons with stroke, though not care partners. Although our hypothesis that depressive symptoms would be reduced in the person with stroke and their care partner was not supported, our results align with other findings that dyadic interventions most often benefit persons with disability rather than their care partners [13,14]. Future work can determine how best to support *both* members of the dyad.

As a pilot study, we may have lacked sufficient power to detect changes in care partner depression scores. Additionally, although some variability existed in care partner depression scores, most were not depressed, and thus, there may have been a floor effect. Care partners may have joined the intervention to support their partner rather than help their own mild depression, thus taking on the responsibility of encouraging the individual with stroke, rather than focusing on their own needs. Additional evidence for a floor effect comes from our exploratory analyses showing that those participants who had more severe depressive symptoms received the most benefit in our study. There could also be referral bias, with clinicians more likely to suggest ReStoreD enrollment to dyads in which the patient was depressed; often, clinicians are less aware of care partner health status or needs [40,41]. There also may have been changes in psychosocial factors not captured by our measures, such as relationship satisfaction, self-efficacy, or social support network.

While care partners may not have directly benefited from intervention activities, our APIM analyses showed a possible negative relationship between depressive symptoms in the person with stroke pre-intervention and the care partner’s depressive symptoms at the 3-month follow-up. This suggests that more depressive symptoms in the partner with stroke at the start of the intervention predicted fewer depressive symptoms in the care partner 3 months after ending the intervention. One potential interpretation of this finding is that care partners may have felt that they were doing something to support their partner who was struggling with post-stroke depressive symptoms, which may have helped them with their own depressive symptoms by the end of the study. Previous work has shown that resilience and self-efficacy are positively associated with stroke care partner quality of life [42]. Another possibility is that those individuals with stroke who had the highest pre-intervention depression scores improved the most with the intervention, and this improvement may be driving improvement in the depressive symptoms of the care partner at follow-up. Based on participant feedback from the study, it is also possible that care partners may have felt validated through the intervention process and gained a better understanding of post-stroke changes, improving their own mood.

### Study Limitations and Future Directions

The primary limitation of our study was the attrition from our waitlist control group, leading to our small sample size. The pilot study was not powered for small-to-medium effects sizes, but it serves to demonstrate proof-of-concept. Additionally, while at least one dyad member was required to report depressive symptoms for study inclusion, the relatively low levels of depression, particularly among care partners, may have limited the ability of the intervention to show a benefit. Finally, this study only included individuals who were interested in participating. Individuals who self-select into these types of studies are likely different (perhaps less depressed and more motivated) from those who do not. Future iterations may specifically target those individuals with more severe depressive symptoms. We may also impose additional structure to enhance both partners’ ability to engage in intervention activities to lighten the responsibility of the care partner to organize, plan, and lead activities. A study using a larger, more diverse sample is needed to determine the efficacy of the intervention for the broader population and may allow us to further explore both care partner and interactive effects within the couple.

## 5. Conclusions

Post-stroke depression is complex and not well understood; currently, there are no proven treatments. Although not all persons with stroke develop mental health disorders or pathology, mental health post-stroke should be a focus of practice across the care spectrum. It is important to consider not only the patient’s mental health but also that of the care partner and the context of the couple’s relationship. This study demonstrates the potential benefits of a novel intervention to treat post-stroke depressive symptoms and lays the groundwork for future research in this area. We establish proof-of-concept by demonstrating that depressive symptoms can be lessened in persons with stroke and care partners with more significant depressive symptoms. The current study’s findings emphasize the importance of future research to establish the efficacy of the ReStoreD intervention in a fully powered study.

## Figures and Tables

**Figure 1 ijerph-19-01804-f001:**
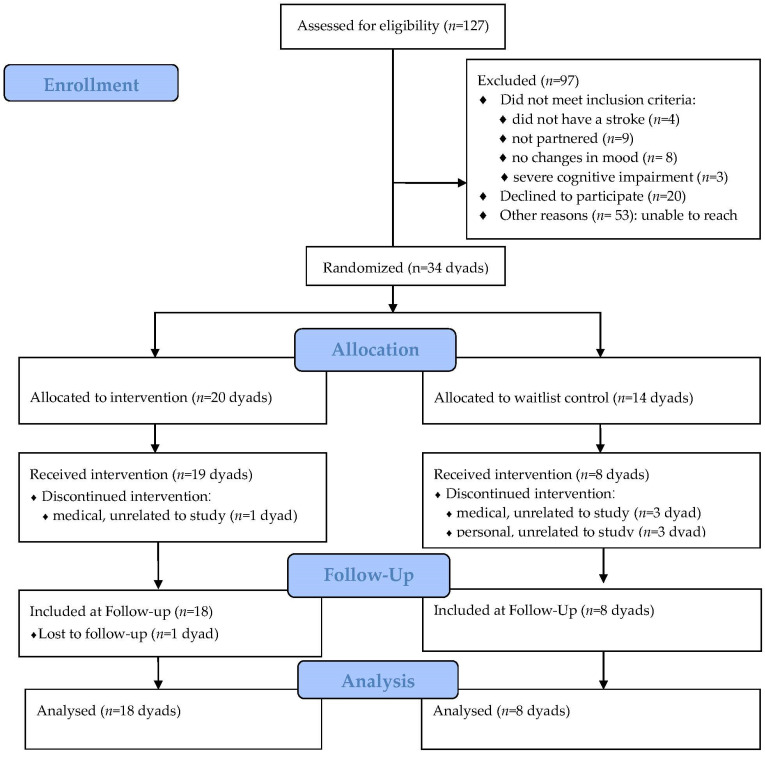
CONSORT diagram.

**Figure 2 ijerph-19-01804-f002:**
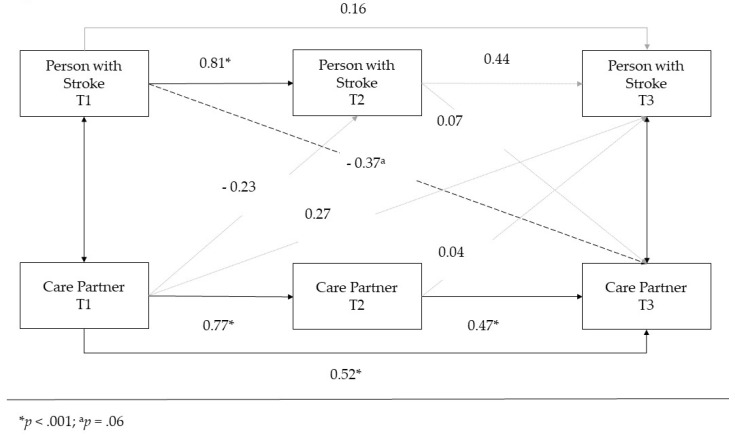
Simplified APIM path model.

**Table 1 ijerph-19-01804-t001:** ReStoreD activities, descriptions, and examples.

Activities	Descriptions	Examples *
Gratitude	Be grateful for life circumstances and persons.	Write a thank you note to the therapist.
Acts of kindness	Perform good deeds for others.	Drop off a meal for a neighbor who recently had a baby.
Relationships	Strengthen relationships, make time for people and be supportive	Have a family game night without electronic “gadgets”.
Positive focus	Replay positive experiences	Tell the partner about progress made during therapy.
Savoring	Replay life’s momentary pleasures, relish ordinary experiences	Watch the sunset together.
Goals	Identify a meaningful goal and devote time to pursuing it	Cook more often/eat out less.
Finding meaning	Seek meaning and purpose, find the sacred in ordinary life	Sharing life goals with the partner.

* All examples were provided by participants in this study.

**Table 2 ijerph-19-01804-t002:** Participant characteristics at enrollment.

Individual Characteristics	Partners with Stroke (*n* = 34)	Care Partners (*n* = 34)
Female, *n* (%)	14 (41.17)	20 (58.82)
Age, mean years (SD)	53.37 (16.14)	52.97 (14.38)
Married, *n* (%)	62 (91.2)	
Length of relationship, mean years (SD)	24.89 (17.79)	
Education/Employment		
>12 years of education, *n* (%)	26 (76.47)	27 (79.41)
Full- or part-time work, *n* (%)	3 (8.82)	19 (55.88)
Non-paid work (e.g., homemaker), *n* (%)	6 (17.65)	2 (5.88)
Retired, *n* (%)	16 (47.06)	11 (32.35)
Unemployed, *n* (%)	8 (23.53)	1 (2.94)
Race/Ethnicity		
White, *n* (%)	31 (91.18)	31 (91.18)
Asian, *n* (%)	1 (2.94)	--
Native Hawaiian or Pacific Islander, *n* (%)	--	1 (2.94)
Preferred not to answer/missing, *n* (%)	2 (5.88)	2 (5.88)
Depressive Symptoms		
PROMIS-D-SF, mean raw score (SD)	17.09 (6.93) ^a^	14.12 (5.53) ^a^
Female: PROMIS-D-SF, mean raw score (SD)	16.92 (6.65) ^b^	15.65 (6.25) *
Male: PROMIS-D-SF, mean raw score (SD)	17.20 (7.27) ^b^	11.77 (3.14) *
Taking antidepressants, *n* (%)	14 (43.8)	7 (21.9)
Stroke Characteristics		
Time since stroke, mean years (SD)	3.45 (4.72)	
Stroke type: ischemic, *n* (%)	24 (70.59)	
Stroke location:		
Left hemisphere, *n* (%)	17 (50.00)	
Right hemisphere, *n* (%)	11 (32.35)	
Other (e.g., brainstem, bilateral), *n* (%)	6 (17.65)	
Cognitive Screening Score ^c^		
MoCA, mean score (SD)	16.90 (2.40)	
CASP (*n* = 3), mean score (SD)	32.00 (1.32)	
Physical Function ^d^		
Lawton (IADL) Scale, mean score (SD)	18.83 (4.44)	

* Male vs. female care partner PROMIS-D-SF: *t*(31) = −2.36, *p* = 0.02. ^a^ Person with stroke vs. care partner PROMIS-D-SF: *t*(64) = 1.92, *p* = 0.06. ^b^ Male vs. female person with stroke PROMIS-D-SF: *t*(31) = 0.11, *p* = 0.91. ^c^ MoCA: administered without visual items; maximum score = 22, below 18/22 is typically used as the cut-off for concern for cognitive challenges; CASP: maximum score = 36, with 30/36 typically used as the cut-off for concern for cognitive challenges. Higher scores indicate better cognitive functioning. ^d^ Lawton IADL Scale: maximum score = 24; higher scores indicate better function.

**Table 3 ijerph-19-01804-t003:** Depression scores (PROMIS-D-SF) over time.

Dyad Member	Pre-Intervention (T1) M (SD)	Post-Intervention (T2) M (SD)	Mean Difference (T1–T2)	3-Month Follow-Up (T3) M (SD)	Mean Difference (T2–T3)
Person with stroke					
All	16.68 (6.91)	14.72 (6.40)	1.96; *p* < 0.05	15.40 (6.78)	−0.68, *p* = n/s
Score >50th %ile	22.00 (5.40)	18.31 (7.03)	3.69, *p* < 0.05	19.38 (6.71)	−1.08, *p* = n/s
Care partner					
All	14.50 (5.78)	14.54 (6.65)	−0.04, *p* = n/s	14.41 (6.23)	0.13, *p* = n/s
Score >50th %ile	18.54 (4.93)	17.23 (7.21)	1.31, *p* = n/s	17.08 (6.90)	0.15, *p* = n/s

Note: M = mean; SD = standard deviation; n/s= not significant, *p* > 0.05.

**Table 4 ijerph-19-01804-t004:** Direct, specific indirect, and total effects for APIM analysis.

Path	B (SD)	[95% CI]
T1 Stroke → T2 Stroke	0.81 (0.15) *	[0.51–1.10]
T1 Partner → T2 Stroke	−0.23 (0.16)	[−0.55–0.16]
T1 Partner → T2 Partner	0.77 (0.22) *	[0.23–1.17]
T1 Stroke → T2 Partner	−0.06 (0.17)	[−0.47–0.23]
T2 Partner → T3 Stroke	0.04 (0.24)	[−0.54–0.47]
T2 Stroke → T3 Stroke	0.44 (0.35)	[−0.37–1.08]
T2 Partner → T3 Partner	0.47 (0.15) *	[0.19–0.75]
T2 Stroke → T3 Partner	0.07 (0.22)	[−0.34–0.49]
**Total Actor Effects: T1 Stroke → T3 Stroke**	0.54 (0.23) ***	[0.04–0.95]
*Direct Effect: T1 Stroke → T3 Stroke*	0.16 (0.34)	[−0.50–0.84]
*Indirect Effects Total*	0.36 (0.28)	[−0.23–0.93]
T1 Stroke → T2 Partner → T3 Stroke	0.01 (0.04)	[−0.10–0.10]
T1 Stroke → T2 Stroke → T3 Stroke	0.37 (0.29)	[−0.25–0.94]
**Total Actor Effects: T1 Partner → T3 Partner**	0.85 (0.18) *	[0.48–1.17]
*Direct Effect: T1 Partner → T3 Partner*	0.52 (0.18) **	[0.15–0.83]
*Indirect Effects Total*	0.32 (0.17) ***	[0.04–0.67]
T1 Partner → T2 Partner → T3 Partner	0.33 (0.15)	[0.09–0.65]
T1 Partner → T2 Stroke → T3 Partner	−0.01 (0.06)	[−0.17–0.10]
**Total Partner Effects: T1 Stroke → T3 Partner**	−0.33 (0.13) **	[−0.63–−0.13]
*Direct Effect: T1 Stroke → T3 Partner*	−0.37 (0.21) ^a^	[−0.81–0.01]
*Indirect Effects Total*	0.03 (0.19)	[−0.37–0.44]
T1 Stroke → T2 Partner → T3 Partner	−0.3 (0.08)	[−0.24–0.10]
T1 Stroke → T2 Stroke → T3 Partner	0.06 (0.18)	[−0.29–0.39]
**Total Partner Effects: T1 Partner → T3 Stroke**	0.18 (0.28)	[−0.40–0.72]
*Direct Effect: T1 Partner → T3 Stroke*	0.27 (0.34)	[−0.47–0.98]
*Indirect Effects Total*	−0.07 (0.23)	[−0.66–0.31]
T1 Partner → T2 Partner → T3 Stroke	0.03 (0.19)	[−0.41–0.36]
T1 Partner → T2 Stroke → T3 Stroke	−0.08 (0.13)	[−0.44–0.11]

Note: T1 = pre-intervention; T2 = post-intervention; T3 = 3-month follow-up. * *p* < 0.001; ** *p* < 0.005; *** *p* < 0.01; ^a^
*p* < 0.05.

## Data Availability

The study was pre-registered at clinicaltrials.gov (NCT#03335358).

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
