# Peer review of "Intimate Relationships and Stroke: Piloting a Dyadic Intervention to Improve Depression"

_ijerph, 2022, doi:10.3390/ijerph19031804_

Round 1

Reviewer 1 Report

This is a very interesting and well written paper. Depression is a complex and multi factorial disease affecting a high percentage of patients with chronic disease.

I have a few comments.

  1. In the analysis of stroke in table 2 a description of disability physical,  cognitive and speech is  very important. Caring and living with a person who cannot move at all or communicate is different from living with a person with minor disability who can speed and socialize. Details of disability should be added to the description .
  2. A second parameter which is crucial in depression analysis is cost. Rehabilitation is crucial for normalizing life but has a high cost. The availability of help at home or at specialized centers, help at home for cleaning, caring or physiotherapy can minimize stress. It would be interesting to know how many coupes had insurance coverage for help at home and rehabilitation facilities. If not, how many had the ability to cover the expenses on their own?
  3. Family. Families tend to go through difficulties easier than lonely people with no support . It would be interesting to know is the depressing couples have a supporting family or children brothers sisters etc.

Author Response

Comment from R1: This is a very interesting and well written paper. Depression is a complex and multi factorial disease affecting a high percentage of patients with chronic disease.

Thank you for your positive response to our manuscript.

  1. In the analysis of stroke in table 2 a description of disability physical,  cognitive and speech is  very important. Caring and living with a person who cannot move at all or communicate is different from living with a person with minor disability who can speed and socialize. Details of disability should be added to the description .

Thank you for this suggestion. We agree that type and severity of these aspects of stroke-related disability are important to consider. Our sample was somewhat restricted in terms of range, as all participants needed to be able to provide their own informed consent, and be able to read, understand, and follow instructions in order to meaningfully participate in the intervention. As such, although we included individuals with mild-moderate cognitive impairment (as assessed using a modified MoCA or CASP as appropriate for those with aphasia) and mild-moderate expressive aphasia, those with severe cognitive impairment and aphasia were excluded. To better describe this range of cognitive impairment in Table 2, we added lines for modified MoCA and CASP scores. To address the physical disability description, we added Lawton Instrumental Activities of Daily Living (IADL) Scale scores (along with interpretation) to the table as a proxy.

 We also added a couple of sentences to the Participant Descriptive data text (lines 178-185):

A range of stroke types and locations were represented among participants with stroke. Consistent with inclusion criteria, individuals with stroke’s cognition screen scores were indicative of a range of impairment from none to moderate. A range of physical function was also represented, as indicated by scores on the Lawton Instrumental Activities of Daily Living (IADL) scale administered pre-intervention (range from 9-24; mean = 18.83, SD = 4.44). “

2. A second parameter which is crucial in depression analysis is cost. Rehabilitation is crucial for normalizing life but has a high cost. The availability of help at home or at specialized centers, help at home for cleaning, caring or physiotherapy can minimize stress. It would be interesting to know how many coupes had insurance coverage for help at home and rehabilitation facilities. If not, how many had the ability to cover the expenses on their own?

We agree that this is in interesting –and important- question. Unfortunately, this was not assessed as part of this study. We will consider including this as part of our upcoming larger trial of the intervention.

3. Family. Families tend to go through difficulties easier than lonely people with no support . It would be interesting to know is the depressing couples have a supporting family or children brothers sisters etc.

We agree that the family context is important to consider as well, as many couples may or may not have support systems outside of the dyad. We did collect some of this data, including how many had children under 18 years living at home (36.4% did), and others who were living in the home (20.6% indicated they had other relatives besides children living in their home; 6.3% had friends living with them). However, we did not collect information on how supportive these are perceived to be and how this relates to depression, which certainly would be an interesting question to consider for future studies.

Reviewer 2 Report

This is a pilot study aimed to test a novel 8-week remotely-administered dyadic intervention (ReStoreD) designed to help couples better cope with stroke-related changes and reduce depressive symptoms. Due to limited sample size, the results did not meet the authors’ expectation. However, the findings provide the proof-of-concept for the future research. This is a well-written manuscript. There are some comments below.

  1. Please provide ClinicalTrials.gov Identifier number.
  2. In Figure 1, it is better to depict the remaining number of people in each step and show the number of people of each exclusion criteria. In addition, it is quite confusing about the meaning of n=0. Please revised it.
  3. Suggested to show p-value in Table 2 and Table 3 instead of in the manuscript. It is much better for the readers to understand.
  4. The study might be biased by volunteer or subjects who had more incentive or willing. How to prevent the bias? The authors should discuss it.

Author Response

Comment Reviewer 2: This is a pilot study aimed to test a novel 8-week remotely-administered dyadic intervention (ReStoreD) designed to help couples better cope with stroke-related changes and reduce depressive symptoms. Due to limited sample size, the results did not meet the authors’ expectation. However, the findings provide the proof-of-concept for the future research. This is a well-written manuscript. There are some comments below.

 Thank you for your positive feedback.

  1. Please provide ClinicalTrials.gov Identifier number.

We have provided the ClinicalTrials.gov in the Data Availability Statement of the manuscript: “The study was pre-registered at clinicaltrials.gov (NCT#03335358).”

In addition, we have also added the following statement to line 89 in Methods: “The study was pre-registered at ClinicalTrials.gov (NCT#03335358).”

2. In Figure 1, it is better to depict the remaining number of people in each step and show the number of people of each exclusion criteria. In addition, it is quite confusing about the meaning of n=0. Please revised it.

Thank you for this feedback. We have revised Figure 1; to include: specifically showing how many participant dyads were allocated to each group, as well as how many completed the intervention as intended, and how many were retained at follow-up and included in analysis. We removed the n=0 as suggested. We hope this clarified the Figure for reviewers/readers.

3. Suggested to show p-value in Table 2 and Table 3 instead of in the manuscript. It is much better for the readers to understand.

We appreciate the suggestion and have addressed it to our best ability as follows: For Table 2, because we are showing differences between partners and persons with stroke as well as the gender interaction (i.e., men vs women who are partners vs persons with strokes), displaying the p-values in the table gets messy, and we chose to provide the p-value information just below the table instead.

We changed Table 3 to be in line with the reviewer’s suggestion, adding columns into the table to show mean differences as well as p-values for better readability. We hope this satisfies the reviewer’s request.

4. The study might be biased by volunteer or subjects who had more incentive or willing. How to prevent the bias? The authors should discuss it.

We agree with the reviewer that this is a potential limitation, and included a brief discussion in the Study Limitation and Future Directions (lines 292-295) section:

“Finally, this study only includes individuals who were interested in participating. Individuals who self-select into these type of studies are likely different (perhaps less depressed, and more motivated) than those who do not.”